# Oxidative Damage in Sporadic Colorectal Cancer: Molecular Mapping of Base Excision Repair Glycosylases MUTYH and hOGG1 in Colorectal Cancer Patients

**DOI:** 10.3390/ijms23105704

**Published:** 2022-05-20

**Authors:** Miriam J. Kavec, Marketa Urbanova, Pavol Makovicky, Alena Opattová, Kristyna Tomasova, Michal Kroupa, Klara Kostovcikova, Anna Siskova, Nazila Navvabi, Michaela Schneiderova, Veronika Vymetalkova, Ludmila Vodickova, Pavel Vodicka

**Affiliations:** 1Department of Molecular Biology of Cancer, Institute of Experimental Medicine of the Czech Academy of Sciences, Videnska 1083, 142 00 Prague, Czech Republic; miriam.kavec@iem.cas.cz (M.J.K.); alena.opattova@iem.cas.cz (A.O.); kristyna.tomasova@iem.cas.cz (K.T.); michal.kroupa@iem.cas.cz (M.K.); anna.siskova@iem.cas.cz (A.S.); nazila.navvabi@gmail.com (N.N.); veronika.vymetalkova@iem.cas.cz (V.V.); ludmila.vodickova@iem.cas.cz (L.V.); 2Department of Oncology, First Faculty of Medicine, Charles University and Thomayer Hospital, 140 59 Prague, Czech Republic; 3Institute of Biology and Medical Genetics, First Faculty of Medicine, Charles University, Albertov 4, 128 00 Prague, Czech Republic; marketku@gmail.com; 4Department of Biology, Faculty of Education, J Selye University, Bratislavska 3322, 945 01 Komarno, Slovakia; makovickyp@ujs.sk; 5Biomedical Centre, Faculty of Medicine in Pilsen, Charles University, Alej Svobody 1655, 323 00 Pilsen, Czech Republic; 6Laboratory of Cellular and Molecular Immunology, Institute of Microbiology of the Czech Academy of Sciences, Videnska 1083, 142 20 Prague, Czech Republic; klimesov@biomed.cas.cz; 7Department of Surgery, General University Hospital in Prague, First Medical Faculty, Charles University, Katerinska 1660, 128 00 Prague, Czech Republic; michaela.schneiderova@vfn.cz

**Keywords:** Oxidative DNA damage, DNA repair, BER glycosylases, colorectal cancer

## Abstract

Oxidative stress, oxidative DNA damage and resulting mutations play a role in colorectal carcinogenesis. Impaired equilibrium between DNA damage formation, antioxidant status, and DNA repair capacity is responsible for the accumulation of genetic mutations and genomic instability. The lesion-specific DNA glycosylases, e.g., hOGG1 and MUTYH, initiate the repair of oxidative DNA damage. Hereditary syndromes (*MUTYH*-associated polyposis, *NTHL1*-associated tumor syndrome) with germline mutations causing a loss-of-function in base excision repair glycosylases, serve as straight forward evidence on the role of oxidative DNA damage and its repair. Altered or inhibited function of above glycosylases result in an accumulation of oxidative DNA damage and contribute to the adenoma-adenocarcinoma transition. Oxidative DNA damage, unless repaired, often gives rise G:C > T:A mutations in tumor suppressor genes and proto-oncogenes with subsequent occurrence of chromosomal copy-neutral loss of heterozygosity. For instance, G>T transversions in position c.34 of a *KRAS* gene serves as a pre-screening tool for *MUTYH*-associated polyposis diagnosis. Since sporadic colorectal cancer represents more complex and heterogenous disease, the situation is more complicated. In the present study we focused on the roles of base excision repair glycosylases (hOGG1, MUTYH) in colorectal cancer patients by investigating tumor and adjacent mucosa tissues. Although we found downregulation of both glycosylases and significantly lower expression of hOGG1 in tumor tissues, accompanied with G>T mutations in *KRAS* gene, oxidative DNA damage and its repair cannot solely explain the onset of sporadic colorectal cancer. In this respect, other factors (especially microenvironment) per se or in combination with oxidative DNA damage warrant further attention. Base excision repair characteristics determined in colorectal cancer tissues and their association with disease prognosis have been discussed as well.

## 1. Introduction

Colorectal cancer (CRC) still represents significant social and health problems, predominantly in the developed countries worldwide. As recently reviewed, there were 1,148,515 new colon cancer cases and 576,858 deaths in 2020, whereas rectal cancer was less frequent with 732,210 newly diagnosed cases and 339,022 deaths [1]. Major risk factors in sporadic CRC comprise dietary and lifestyle habits and age [2,3]. Murphy, et al. summarized recently convincing evidence on the role of inflammation, lipid peroxidation, oxidative stress and metabolic dysfunction in the onset of CRC and its development [3]. The formation of reactive oxygen species (ROS) and subsequent DNA damage is unambiguously associated with physiological and pathological processes, such as obesity, diabetes, inflammatory bowel diseases. These processes are known as factors involved in CRC aetiology [4]. For instance, promoted oxidative stress and suppressed immune system are linked with increases in inflammatory factors and adipokines (TNF, leptin, IL-1β, and IL-6) in obesity. Above alterations may result in aberrant cell signaling, increased cell growth and angiogenesis [5,6]. The formation of ROS may also be induced by intestinal bacteria in colonic epithelium [7].

Altered homeostasis in DNA damage levels, capacity for DNA repair, and antioxidant status substantially affect the accumulation of mutations and genomic instability. Once ROS reaches DNA, the oxidation of nucleophilic DNA bases and the ribose sugar ring leads to base loss and strand breaks. The majority of interactions with ROS targets guanine, giving rise to 8-oxo-7,8-dihydro-2’deoxyguanosine (8-oxo-dG) and 2,6-diamino-4-hydroxy-5-formamidopyrimidine (FAPY). Reaction of ROS also proceeds via adenine (8-oxo-7,8-dihydro-2´deoxyadenosine, 2-hydroxyadenine), and to a lesser extent with thymine and cytosine. Unrepaired 8-oxo-dG adducts induce G>T transversions [8,9], in proto-oncogenes, such as *KRAS* gene, and tumor suppressor genes. The main mechanism involved in the removal and repair of oxidized DNA bases with the involvement of lesion-specific DNA glycosylases is base excision repair (BER) [10,11,12]. Additionally, both *MUTYH* and *hOGG1* were found to be downregulated in malignant human colon tissues in comparison with adjacent tissues [13]. The role of BER glycosylases in the transition from early adenoma to CRC has clearly been documented on hereditary syndromes *MUTYH*-associated polyposis (MAP) and *NTHL1*-associated tumor syndrome (NATS) [14,15]. In this context, G>T transversions in position c.34 of a *KRAS* gene was suggested as a pre-screening tool for MAP diagnosis [16]. Individuals and/or cells with deficient DNA glycosylase functions exhibit increased levels of base damage in their DNA, elevated mutation rates, and hypersensitivity to specific DNA damaging agents [17]. As shown by Helleday, et al., somatic mutations (base substitutions, insertions, and deletions or structural variations) in a sporadic cancer represent the outcome of multiple mutagenic steps, determined by the type of DNA damage and their repair processes, over the lifetime of a patient [18]. The suboptimal function of BER glycosylases contributes significantly to these processes.

Here, we provide molecular characteristics of two major BER glycosylases MUTYH and hOGG1 in tissue samples of sporadic CRC patients in relation to a characteristic mutational signature (G:C > T:A transversion in oncogenic *KRAS* gene) with specific aim to assess the role of BER glycosylases on the pathogenesis of sporadic CRC.

## 2. Materials and Methods

### 2.1. Study Population

One hundred and ninety-three newly diagnosed sporadic CRC patients from region of Prague were enrolled in the study since 2010; tumor tissues, adjacent mucosa, and clinicopathological data were available. The study group consisted of 122 males and 71 females with a median age of 69.5 years (range 38–91, mean 68.7). The group of patients encompassed 85 non-smokers, 29 smokers and 79 ex-smokers. The MSI-high phenotype was found in 16 (8.3%) tumor samples, whereas the remaining 177 tumors (91.7%) had microsatellite stable (MSS) phenotype. Regarding the UICC staging, 45 patients were with TNM I, 57 with TNM II, 58 with TNM III and 34 patients with TNM IV. Most tumors were localized in colon (131; 67.5%), followed by rectosigmoideum (19; 9.8%) and rectum (44; 22.7%). The studied population is characterized in Table 1. Study was approved by Ethical committees of First Medical Faculty (12/11 Grant GACR 1. LFUK) and Institute of Clinical and Experimental Medicine (622/11 G11-04-09).

### 2.2. Screening for Mutations and Gene Variants

DNA from tumor and adjacent non-malignant mucosa, extracted using the DNeasy Blood and Tissue Kit (Qiagen, Courtaboeuf, France), was used for amplification of all coding regions and adjacent intron sites of *MUTYH* gene to obtain approximately 200 bp-long fragments. We used 5× HOT FIREPol^®^ EvaGreen^®^ HRM Mix chemistry (Solis Biodyne, Tartu, Estonia) and LightCycler^®^ 480 Instrument (Roche, Indianapolis, IN, USA) to conduct prescreening of amplicons by high resolution melting analysis. Each amplicon melting was precisely optimized. All samples with non-standard melting curves were subsequently sequenced by a modified Sanger method.

SNaPshot analysis was used to record the most prevalent *KRAS* somatic mutations in codon 12 and 13 at nucleotides c.34, 35, 37 and 38 (NM_004985.5) which should comprise 97% of all possible variants [19]. Four different primer pairs were run in a multiplex PCR and followed by the addition of one fluorescently labeled dideoxynucleotide; obtained fragment was put in an ABI 3130 (Applied Biosystems, Waltham, MA, USA) to record all possible alleles and their quantity.

Gene expression of *hOGG1* and *MUTYH* genes was carried out essentially as described by [20]. RNA was isolated using TRIzol (Invitrogen, Waltham, MA, USA) and RNA integrity ranged between 8.0 and 10.0. cDNA was synthesized from 0.5 µg of RNA using the RevertAld^TM^ First-strand cDNA synthesis kit (Fermentas, Burlington, ON, Canada). cDNA was preamplified and qPCR performed using the BioMark^TM^ HD System (Fluidigm, San Francisco, CA, USA). *TOP1* served as a reference gene selected by Normfinder. Normality distribution was tested by Kolmogorov-Smirnov test. According to Kolmogorov-Smirnov test normal distribution was not approved, therefore for the further statistical analysis we used the non-parametrical statistical analysis (Mann-Whitney test). All data set are showed as box plots with 5–95th percentile (Whiskers).

Five repeat markers (Bethesda consensus panel, BAT 25, BAT 26, NR 21, NR 24, and NR 27) that were run as a pentaplex using fluorescently labeled primers, were employed for the determination of MSI status. Fragment analysis was conducted on ABI 3130 (Applied Biosystems, Waltham, MA, USA). GeneMapper v4.1 software (Applied Biosystems, Waltham, MA, USA) was employed to discern a difference between tumor and non-tumor DNA short tandem repetition profiles. Tumor specimen was classified as MSI high when two or more loci were unstable.

For immunohistochemical determination (IHC) of hOGG1 protein the samples were fixed for 24 h with a 10% formalin solution and subsequently processed by standard histological methods. Three to five μm-thick slices were cut from each sample to the special slides (DAKO, Glostrup, Denmark). The first slices were stained with haematoxylin-eosin (DiaPath, Martinengo, Italy) and the second slices with 8-hydroxyguanine DNA glycosylase (OGG1) polyclonal antibody (PA-116505, Thermo Fisher Scientific, Waltham, MA, USA). Before immunostaining, heat-induced antigen retrieval was performed for 20 min. in pH 6.0 citrate buffer (Target Retrieval Solution, Low pH, DAKO, Glostrup, Denmark). The slices were subsequently incubated with the OGG1 antibody (1:100 dilution; Dako Antibody Diluent, DAKO, Glostrup, Denmark) at room temperature for 1 h. Slices were washed in conventional wash buffer (DAKO, Glostrup, Denmark) and visualized using the LSAB+System HRP kit (streptavidin-biotin peroxidase detection kit, DAKO, Glostrup, Denmark). The samples were evaluated using an optical microscope Olympus AX70 Provis (Olympus, Tokyo, Japan).

On this cohort we have also analyzed overall (OS) and event-free survival (EFS). The time that has elapsed from the surgery to the date of death, or the date of last follow-up (November 2019 for this cohort) was assigned as OS. The time from surgery to signs of distant metastasis, local recurrence, or death, whichever appeared first, was defined as EFS. The Kaplan–Meier log-rank test was used to derive the survival curves for OS and EFS. The Cox regression [21] enabled to estimate the relative risk of death and recurrence (expressed as hazard ratio (HR) with 95% CIs). Detailed description of the employed statistical analyses is stated elsewhere [21,22].

## 3. Results

A. Gene variants, somatic mutations and LOH in the *MUTYH* gene.

Interestingly, only 3 somatic substitutions c.38C>T (p.A13V) [rs587780747], c.141G>A (p.K47K) and c.695C>T (p.T232I) in *MUTYH* gene (NM_001128425) have been detected in the tumor tissue; however, with no unambiguous pathological effect on the protein. Predominant germline variants, identified by us, were single nucleotide polymorphisms and 2 germline pathogenic mutations (p. G396N, p.Y104X; Table 2). Regarding the loss of heterozygosity (LOH), we identified 9 samples (4.7%) with at least a partial loss of the *MUTYH* gene based on the SNP data evaluation. In summary, all mutations were found in one allele only and none of the patients exhibited loss of function mutation/variant in *MUTYH*.

B. Relative expressions levels of *MUTYH* and *hOGG1* gene. 

The relative gene expression of both glycosylases was significantly higher in non-malignant mucosa of 91 CRC patients (i.e., in samples with appropriate RNA quality) than in corresponding tumor tissue (Figure 1; *p* = 0.0038 for *MUTYH* and *p* = 0.0016 for *hOGG1*, Mann-Whitney U-test). Relative gene expressions of *MUTYH* and *hOGG1* glycosylases significantly correlated in both tumor (rho = 0.599, *p* ˂ 0.001) and adjacent mucosa (rho = 0.638, *p* ˂ 0.001) tissues (Figure 2A,B). An inverse relationship between *hOGG1* expression and TNM staging was also observed (χ^2^ test, *p* = 0.029). The difference was particularly pronounced by comparing TNM I+II versus TNM III+IV, suggesting that decrease in *hOGG1* expression may be associated with increasing tumor progression. The relative expression of *MUTYH* gene in tumor did not differ in relation to *MUTYH* gene alterations (somatic and germline mutations, LOH). *MUTYH* gene expression was not associated either with TNM status or with the age at diagnosis.

Relative expression of *KRAS* was almost identical in both tumor tissues and adjacent mucosa (data not shown). A lack of difference in gene expression does not preclude activation of this oncogene.

C. hOGG1 protein expression by immunohistochemistry. 

Protein expression of BER glycosylase *hOGG1* was characterized by IHC on a limited set of CRC patients (*n* = 39). IHC was determined in epithelium and stroma of non-affected mucosa, tumor tissue, epithelium adjacent to tumor and stroma adjacent to tumor. Figure 3A illustrates the highest expression of hOGG1 protein in epithelial cells (both adjacent to tumor and non-affected one), and the lowest in stroma corresponding to non-affected mucosa and tumor. Importantly, the hOGG1 expression was significantly lower in malignant cells and accounted for about half of the expression in non-malignant epithelium. Despite the scarcity of data prevented correlation between gene and protein expressions of *hOGG1* glycosylase, both gene and protein expressions suggest lower levels in tumor tissues. Figure 3(B1) represents IHC slide showing surrounding epithelial cells, adenocarcinoma cells and stromal cells. Figure 3(B2) illustrates IHC staining of non-affected mucosa with positive epithelial cells and less positive stromal cells.

D. *KRAS* somatic mutations at positions c.34, 35, 37 and 38. 

By mapping oncogenic *KRAS* mutations in sporadic CRC patients, we found 51 *KRAS* mutations in 50 patients (29.5%). Most substitutions were in position c.35 (74.5%), namely c.35 G>A (39.2%) followed by c.35 G>T (25.5%; Figure 4). Transversion c.34 G>T, suggested as a prescreening tool for MAP patients [16], was present in only 2 patients, one of them with monoallelic germline nonsense substitution in *MUTYH* gene (p.Tyr104X). Our main interest was substitution G>T in any of the *KRAS* hotspot positions as a possible consequence attributable to the main oxidative DNA damage, 8-OH-G. In total we found 15 of these transversions in *KRAS* positive tumor DNA (30.0%), which stands for 7.8% of all tumor DNA in our set. Interestingly, we found 31 G>A transitions, which may be associated with oxidized purines, oxidized adenines or alkylations at O^6^-position of guanine.

Regarding the UICC TNM staging, there was no significant association between staging and *KRAS* mutations (χ^2^ test, *p* = 0.55). Neither were there any associations between *KRAS* mutations and grade of dysplasia (χ^2^ test, *p* = 0.51) or age at diagnosis (χ^2^ test, *p* = 0.26). Among patients with MSI-high CRC phenotype (*n* = 16, 8.3% of all patients) we did not record any *KRAS* mutations.

E. Survival analysis.

The follow-up for both OS and EFS was carried out between the beginning of year 2010 and November 2019. At the end of the follow-up 85 patients died and 108 were alive. Both OS and RFS were shorter with increasing age (log-rank test; *p* = 0.0005 and *p* = 0.006, respectively). In our set of CRC patients, OS and RFS were strongly affected by the TNM status (log-rank test; *p* < 0.0001 for both) and grade of dysplasia (log-rank test; *p* = 0.013 and *p* = 0.004, respectively). However, neither OS nor EFS were associated with sex, smoking habit, the KRAS positivity, the *MUTYH* gene sequence aberrations and the level of gene expression of *MUTYH* and *hOGG1* and these findings conform with those in TCGA [25].

## 4. Discussion

There is an indisputable role of inflammation, lipid peroxidation, oxidative stress and metabolic dysfunction in etiopathogenesis of sporadic CRC (reviewed in [3]). In this context, obesity, diabetes, and inflammatory bowel diseases, known to be involved in CRC development, trigger the formation of reactive oxygen species (ROS) and subsequent DNA damage induction [4]. In obesity, oxidative stress linked with alteration of the immune system and with ultimate aberrant cell signaling, stimulated cell growth and angiogenesis is promoted via enhanced inflammatory factors and adipokines (TNF, leptin, IL-1β, and IL-6) [5,6]. Further, intestinal bacteria in colonic epithelium may also contribute to ROS [7]. Recent studies on hereditary syndromes, such as MAP polyposis, and NTH1 polyposis [15,26], associated with CRC, suggest etiopathogenic role of oxidative DNA damage along with inappropriate function of BER glycosylases in colorectal carcinogenesis. In sporadic CRC, a generation of oxidative DNA damage is also associated with intestinal microenvironment (microbiota) and these significantly modulate immune response. Further, high amounts of ROS are also produced by tumor cells [27]. Whether high levels of oxidative damage in CRC cells emerge in early carcinogenesis or as its consequence remains enigmatic.

The role of DNA damage in carcinogenesis is rather versatile. Whereas its higher extent may trigger malignant transformation very early, in developed/advanced tumors its lower level (or efficient BER) may give additional survival/growth advantage to cancer cells [28]. However, a comparison of BER capacity between the tumor tissues and adjacent mucosa from sporadic CRC patients did not disclose major differences [29]. A recent study by Vodenkova, et al. demonstrated that low BER in tumor and higher BER capacity in adjacent mucosa conferred to significantly longer survival and vice versa. Additionally, higher BER capacity in tumor tissue as compared to adjacent mucosa was significantly associated with advanced tumor stage [22]. Due to their importance, oxidative DNA damage and its repair should be further investigated, especially in relation to the complex diseases. In addition, BER and oxidative DNA damage may participate in a complex response observed in combination therapies (including immunotherapy), as well as they may represent some of many players in acquired resistance towards chemotherapy [30,31,32].

To address some of these points in sporadic CRC patients, we carried out molecular characteristics of two major BER glycosylases in tissue samples of 193 newly diagnosed sporadic CRC patients. We assumed that oxidative DNA damage induces predominantly G>T transversion rather than C > T transition (attributable to oxidative damage in pyrimidines). Regarding BER glycosylase MUTYH, all mutations were found in one allele only and none of the patients exhibited loss of function mutation/variant in *MUTYH*. Relative gene expressions of both *MUTYH* and *hOGG1* were significantly lower in tumor tissue that in adjacent mucosa, so was hOGG1 protein expression determined by IHC. Similar tendency of downregulated *MUTYH* and *hOGG1* in tumor tissues was also found on a group of 49 sporadic CRC patients from Brazil [13]. Interestingly, a good correlation between expressions of *hOGG1* and *MUTYH* genes was found in both tumor tissue and adjacent mucosa. Fine coordination between these BER glycosylases is particularly pronounced in non-malignant mucosa, whereas moderately looser in tumor tissue. Underlying biological significance needs to be verified on larger set of samples, enabling proper stratification for localization, CRC phenotype and TNM. The presence of *KRAS* mutation in our set of patients agreed with that in other populations (29.5% versus 27.4% in UK, 38% in Switzerland, and 41% in Spain). Regarding *KRAS* mutations in *KRAS* positive tumor DNA, G>T transversion in any of the *KRAS* hotspot positions (attributable to the main oxidative DNA damage, 8-oxo-dG) accounted for 30.0%. Transitions G>A represented the majority of mutations. These may be associated with oxidized purines, oxidized adenines or alkylations at O^6^-position of guanine [15,26,33,34]). Although oxidative DNA damage and its repair cannot solely explain the onset of sporadic CRC, their role in combination with other factors deserves further attention. BER characteristics determined in CRC tissues may rather reflect disease prognosis. Importantly, oxidative DNA damage may be repaired from DNA also via NER [35,36]. This implies that the loss of NER function shares common features with BER defects, including cancer predisposition [37]. Gene variants in DNA repair genes may alter DNA repair function, including function of BER glycosylases, modulate its capacity, and induce genetic instability or deregulate cell growth and propagate cancer [38,39,40]. A meta-analysis comprising 4174 cases and 6196 controls did not discover any significant association between *hOGG1* Ser326Cys polymorphism and CRC, however, further investigation was recommended [41,42]. In a meta-analysis (enrolling more than 8000 CRC cases and 6000 controls) Picelli, et al. revisited the associations of rs3219484:G–A (MUTYH V22M) and rs3219489:G-C (MUTYH Q338H) polymorphisms with the risk of sporadic CRC with negative outcomes [43].

Oxidative DNA damage continues to be an important factor in etiopathogenesis of CRC and its further monitoring is therefore warranted. Similarly, it may also represent a significant marker of prognosis and its level may contribute to treatment outcome.

With accumulating knowledge on the role of microenvironment in colorectal carcinogenesis the dynamic of DNA damage formation (and oxidative DNA damage in particular) and its repair should be given proper attention. Despite certain limitations of our study, such as limited patient´s population and a lack of simultaneous analyses of microbiota, oxidative DNA damage and BER, in our cohort of sporadic CRC patients the oxidative DNA damage and BER glycosylases cannot solely explain the onset of sporadic CRC.

## Figures and Tables

**Figure 1 ijms-23-05704-f001:**
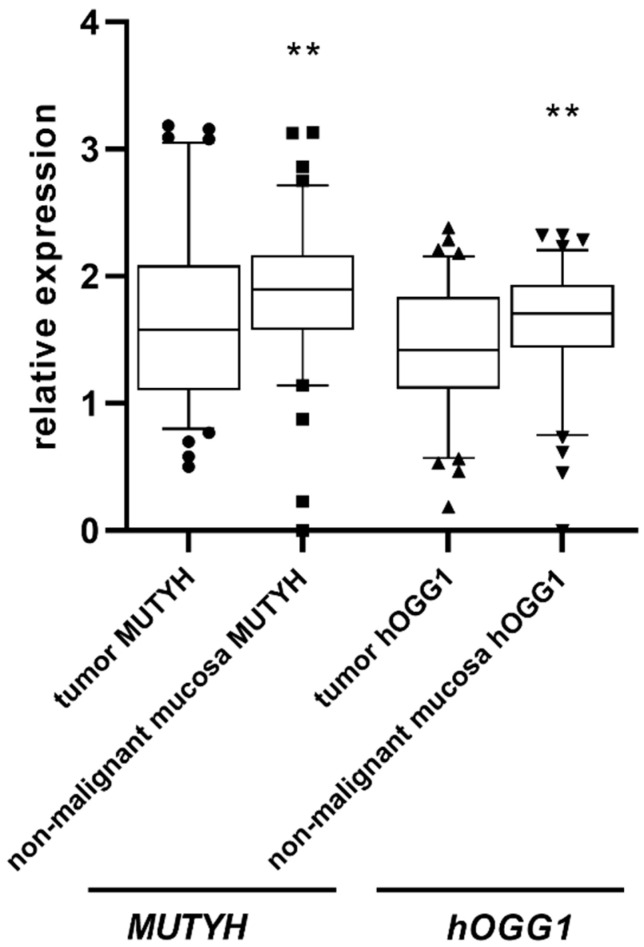
*MUTYH* and *hOGG1* relative gene expression showed higher level in non-malignant mucosa than in corresponding tumor tissue (** *p* = 0.0038 for MUTYH and ** *p* = 0.0016 for hOGG1, all data are showed as median with 5–95th percentile, Mann-Whitney test; *n* = 91).

**Figure 2 ijms-23-05704-f002:**
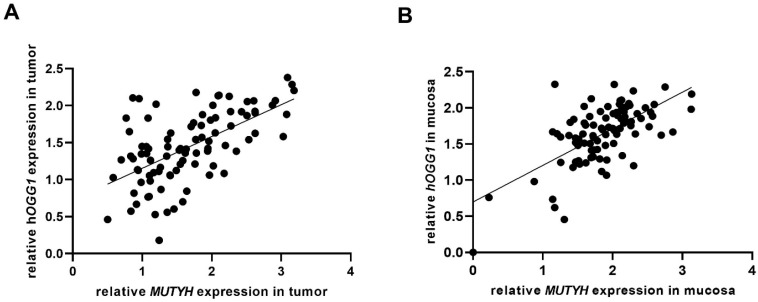
(**A**,**B**). *MUTYH* and *hOGG1* relative gene expressions significantly correlated in both tumor (rho = 0.599, *p* ˂ 0.001) and adjacent mucosa (rho = 0.638, *p* ˂ 0.001; *n* = 91).

**Figure 3 ijms-23-05704-f003:**
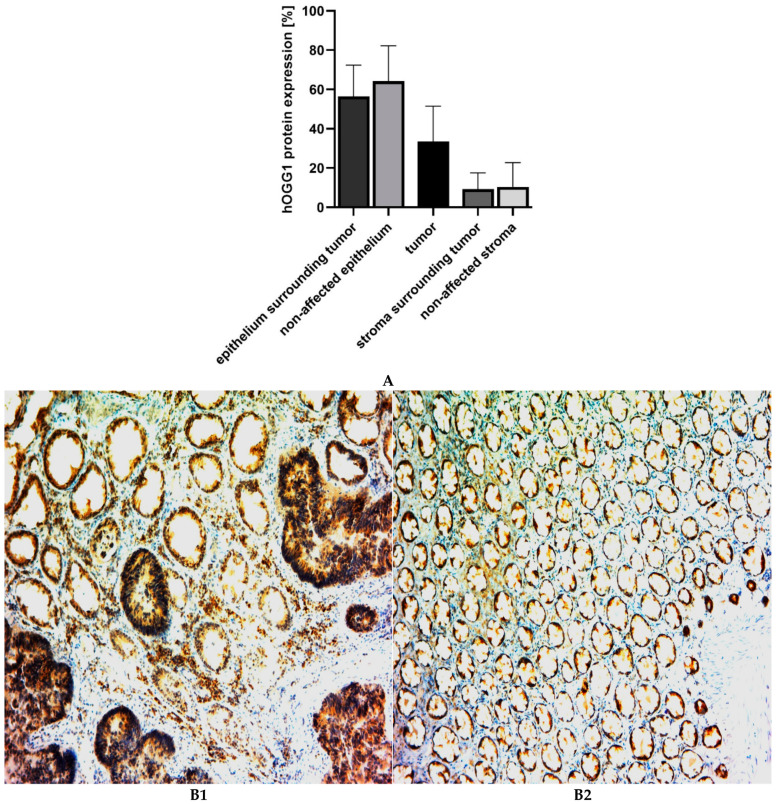
(**A**) hOGG1 protein expression levels in different tissues (*n* = 39). (**B1**) Representative IHC slide showing surrounding epithelial cells, adenocarcinoma cells and stromal cells. Epithelial cells exhibited highest and most consistent positivity of hOGG1, positivity in tumor is lower and the lowest is in stromal cells. (**B2**) IHC slide showing non-affected mucosa with positive epithelial cells and less positive stromal cells.

**Figure 4 ijms-23-05704-f004:**
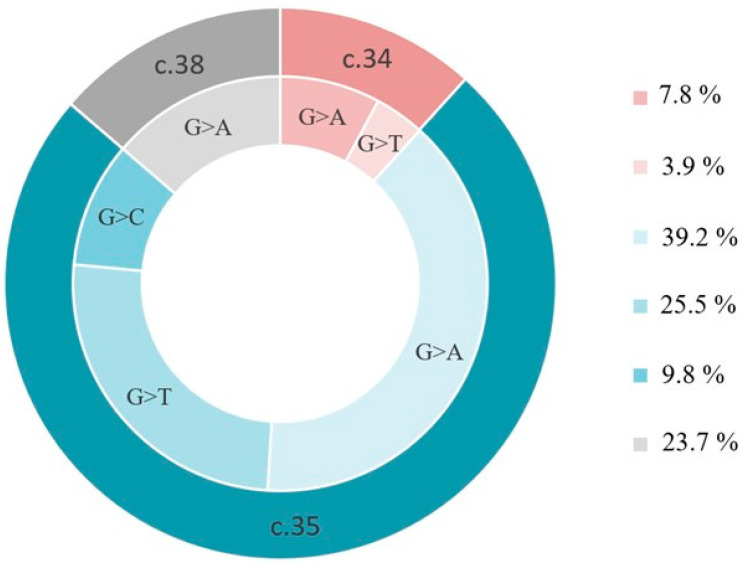
*KRAS* mutations distribution detected in our patient cohort (*n* = 50).

**Table 1 ijms-23-05704-t001:** Characteristics of the studied population.

		Number of Patients (*n* = 193)
Gender	Male	122 (63.2%)
	Female	71 (36.8%)
Age of diagnosis	Median	69.5
	Range	38–91
Smoker	Smoker	29 (15%)
	Non-smoker	85 (44%)
	Ex-smoker	79 (41%)
TNM stage	I	45 (23%)
	II	57 (29.5%)
	II	58 (30%)
	IV	34 (17.5%)
MSI phenotype	MSI-high	16 (8.3%)
	MSI-low (stable)	177 (91.7%)
Localization	colon	131 (67.5%)
	rectosigmoideum	19 (9.8%)
	rectum	44 (22.7%)
Grading (*n* = 64)	NDA	4 (6.25%)
	0	6 (9.38%)
	1	3 (4.68%)
	1–2	6 (9.38%)
	2	36 (56.25%)
	2–3	3 (4.68%)
	3	5 (7.81%)
	4	1 (1.56%)

**Table 2 ijms-23-05704-t002:** Gene variants and somatic mutations in *MUTYH* gene (NM_001128425).

Somatic Mutations			
Carrier/Gender/Age	Gene Position	Protein Position	Reference, rs#
P137/F/60	c.141 G>A	p.Lys47Lys	novel
P157/F/77	c.695 C>T	p.Thr232Ile	novel
P257/M/70	c.38 C>T	p.Ala13Val	rs587780747
**Germline Mutation**			
P145/M/85	c.312 C>A	p.Tyr104X	[23]
P13/F/75	c.1187 G>A	p.Gly396Asp	[24], rs36053993
**Benign Exonic SNPs**			
**Frequency in our Sample Group**	**Gene Position**	**Protein Position**	**SNP ID**
16/193	c.64 G>A	p.Val22Met	rs3219484
1/193	c.312C>T	p.Tyr104Tyr	rs121908380
31/193	c.1014G>C	p.Gln338His	rs3219489
2/193	c.1276C>T	p.Arg426Cys	rs150792276
1/193	c.1431G>C	p.Thr477Thr	novel

## Data Availability

Not applicable.

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
