# Peer review of "Oxidative Damage in Sporadic Colorectal Cancer: Molecular Mapping of Base Excision Repair Glycosylases MUTYH and hOGG1 in Colorectal Cancer Patients"

_ijms, 2022, doi:10.3390/ijms23105704_

Round 1

Reviewer 1 Report

figure 1 is not statistically correct. I strongly doubt the statistics. While figure 2 and 3 may be statistically significant there are no statistics.   figure 3 is IHC but there is no image of IHC..i am not sure what authors are hiding it. Where is the histology grading ? Where is the western data?    Authors either show the data or just don't publish it. 

Author Response

The authors are thankful to the reviewer for his/her valuable comments. We did our best to response all of them and addressed point-by-point all the changes performed in the manuscript (they are highlighted in the manuscript by using a different text colour).

First of all, the manuscript has been adjusted in line to eliminate duplicities/overlaps.

1. Figure 1 is not statistically correct. I strongly doubt the statistics.

We have revisited the original data and these had been recalculated. Normality distribution was tested by Kolmogorov-Smirnov test. According Kolmogorov-Smirnov test normal distribution was not approved, therefore for the further statistical analysis we used the non-parametrical statistical analysis (Mann-Whitney test). All data set are showed as box plots with 5-95th percentile (Whiskers). Since we used non-parametric Mann-Whitney U-test (based on ranking), the overlap of SDs may not reflect a lack of significance.

We have also supplemented the caption for Figure 1.

2. While figure 2 and 3 may be statistically significant there are no statistics.

Figure 2 shows a correlation between the expressions in two compartments. The statistics is shown in the caption, i.e. rho=0,599, P<0,001 for 2A; rho=0,638, P<0,001 for 2B).

For Figure 3 it is a bit meaningless to show a statistics, since very meager protein expression may be expected in stroma. The difference between colonic epithelium and tumor is self-explanatory.

3. Figure 3 is IHC but there is no image of IHC…I am not sure what authors are hiding it.

The reviewer is correct and we have included images for each particular category. The figure 3 has been re-assigned as Figure 3A (histogram) and 3B (images).

4. Where is the histology grading? Where is the western data? Authors either show the data or just don't publish it. 

Regarding the grading, we have added relevant data into the Table 1.

Since all the analyses have been carried out on human biopsies, we performed IHC analyses. No western data were part of the study.

Reviewer 2 Report

An interesting report assessing a timely topic in colorectal cancer and medical oncology in general.

However, we believe some changes are necessary.

  1. A linguistic professional revision is recommended.
  2. The authors should better discuss the limitations of the current report. 
  3. It would be important to include some recent papers discussing the role of base excision repair genes in the current landscape of medical oncology. We recommend including the following papers, only for a matter of consistency (PMID: 34429006; PMID: 35412613).

Author Response

The authors are thankful to the reviewer for his/her valuable comments. We did our best to response all of them and addressed point-by-point all the changes performed in the manuscript (they are highlighted in the manuscript by using a different text colour).

First of all, the manuscript has been adjusted in line to eliminate duplicities/overlaps.

1. A linguistic professional revision is recommended.

We have asked a native English speaker to check our manuscript, at least as carefully as the time allowed.

2. The authors should better discuss the limitations of the current report. 

The main limitations are following: the colorectal cancer heterogeneity is more extensive than we initially thought and the Population size is therefore limited. Another limitation is a lack of simultaneous analyses of microbiota, oxidative DNA damage and BER. These limitations have also been incorporated into discussion part of the manuscript.

3. It would be important to include some recent papers discussing the role of base excision repair genes in the current landscape of medical oncology. We recommend including the following papers, only for a matter of consistency (PMID: 34429006; PMID: 35412613).

Thank you for this suggestions, we have included both references as well as a note to current landscape of medical oncology.

Round 2

Reviewer 1 Report

Figure 1: Just by changing the figure style ; I am not convinced it's really significant . I hope your statistics method is accurate. Put numbers of N in each scientific figure. also you figure to too bright can you add contrast in IHC image. 

However, your science is good.

Author Response

The authors are thankful to the reviewer for his/her valuable comments. We did our best to response all of them and addressed point-by-point all the changes performed in the manuscript (they are highlighted in the manuscript by using a different text colour).

The manuscript has been adjusted in line to eliminate remaining duplicities/overlaps in the line 130-135 and 150-156.

1. Figure 1: Just by changing the figure style; I am not convinced it's really significant. I hope your statistics method is accurate.

By using the non-parametric Mann-Whitney U-test the difference was indeed significant.

2. Put numbers of N in each scientific figure. Also you figure to too bright can you add contrast in IHC image. 

We have added numbers of investigated patients (N) to Figure 1, 2A + 2B, 3A and 4. IHC images (Figure 3B-a and 3B-b) have been adjusted by adding +40 % contrast and removing -20 % brightness. We believe we have improved images according to reviewer’s recommendation.

Reviewer 2 Report

Acceptance.

Author Response

The authors are grateful to the reviewer for his/her valuable comments that have helped to improve the article.